# Different Methods to Improve the Monitoring of Noninvasive Respiratory Support of Patients with Severe Pneumonia/ARDS Due to COVID-19: An Update

**DOI:** 10.3390/jcm11061704

**Published:** 2022-03-19

**Authors:** Paolo Pelosi, Roberto Tonelli, Chiara Torregiani, Elisa Baratella, Marco Confalonieri, Denise Battaglini, Alessandro Marchioni, Paola Confalonieri, Enrico Clini, Francesco Salton, Barbara Ruaro

**Affiliations:** 1Anesthesia and Critical Care, San Martino Policlinico Hospital, IRCCS for Oncology and Neurosciences, 16132 Genoa, Italy; ppelosi@hotmail.com (P.P.); battaglini.denise@gmail.com (D.B.); 2Department of Surgical Sciences and Integrated Diagnostics, University of Genoa, 16132 Genoa, Italy; 3Respiratory Diseases Unit and Center for Rare Lung Disease, Department of Surgical and Medical Sciences SMECHIMAI, University of Modena Reggio Emilia, 41121 Modena, Italy; roberto.tonelli@me.com (R.T.); marchioni.alessandro@unimore.it (A.M.); enrico.clini@unimore.it (E.C.); 4Clinical and Experimental Medicine PhD Program, University of Modena Reggio Emilia, 41121 Modena, Italy; 5Pulmonology Department, Cattinara Hospital, University of Trieste, 34127 Trieste, Italy; torricina@gmail.com (C.T.); marco.confalonieri@asugi.sanita.fvg.it (M.C.); paola.confalonieri.24@gmail.com (P.C.); francesco.salton@gmail.com (F.S.); 6Department of Radiology, Cattinara Hospital, University of Trieste, 34127 Trieste, Italy; elisa.baratella@gmail.com

**Keywords:** COVID-19, respiratory failure, noninvasive respiratory support (NIRS), ARDS, coronavirus disease

## Abstract

The latest guidelines for the hospital care of patients affected by coronavirus disease 2019 (COVID-19)-related acute respiratory failure have moved towards the widely accepted use of noninvasive respiratory support (NIRS) as opposed to early intubation at the pandemic onset. The establishment of severe COVID-19 pneumonia goes through different pathophysiological phases that partially resemble typical acute respiratory distress syndrome (ARDS) and have been categorized into different clinical–radiological phenotypes. These can variably benefit on the application of external positive end-expiratory pressure (PEEP) during noninvasive mechanical ventilation, mainly due to variable levels of lung recruitment ability and lung compliance during different phases of the disease. A growing body of evidence suggests that intense respiratory effort producing excessive negative pleural pressure swings (P_pl_) plays a critical role in the onset and progression of lung and diaphragm damage in patients treated with noninvasive respiratory support. Routine respiratory monitoring is mandatory to avoid the nasty continuation of NIRS in patients who are at higher risk for respiratory deterioration and could benefit from early initiation of invasive mechanical ventilation instead. Here we propose different monitoring methods both in the clinical and experimental settings adapted for this purpose, although further research is required to allow their extensive application in clinical practice. We reviewed the needs and available tools for clinical–physiological monitoring that aims at optimizing the ventilatory management of patients affected by acute respiratory distress syndrome due to severe acute respiratory syndrome coronavirus-2 (SARS-CoV-2) infection.

## 1. Introduction

The coronavirus disease (COVID-19) pandemic, caused by severe acute respiratory syndrome coronavirus-2 (SARS-CoV-2) infection, is still overwhelming global healthcare systems due to the huge spread of a life-threatening pneumonia that as of 25 January 2022 has caused more than 5,832,333 recognized deaths worldwide [1]. The guidelines for the hospital care of patients affected by COVID-19-related acute respiratory failure (ARF) have moved from a prevalent early intubation approach to the widely accepted use of noninvasive respiratory support methods, such as high-flow nasal cannula (HFNC) and noninvasive continuous positive airway pressure (CPAP) and positive pressure ventilation (NIV) [1,2]. This change in the management of the disease has been due to the limited availability and higher cost of intensive care unit (ICU) beds, as well as to the positive results obtained with corticosteroids and noninvasive respiratory support (NIRS) in several published trials [2]. However, the severity of COVID-19 pneumonia and the risk of underestimating the potential dangers of a uselessly prolonged NIRS prompted us to summarize the need for the strict clinical–physiological monitoring of patients affected by severe COVID-19 pneumonia or acute respiratory distress syndrome (ARDS) based on serial imaging techniques and physiological measurements [1,2].

The objective of this narrative review is to discuss the needs and available tools for clinical–physiological respiratory monitoring that aims at optimizing the respiratory assistance and the ventilatory management of patients affected by ARDS due to SARS-CoV-2 infection.

## 2. Physio-Pathological Differences and Similarities between COVID-19 and Non-COVID-19 ARDS

Both COVID-19 and non-COVID-19 ARDS are characterized by severe refractory hypoxemia and high mortality [3,4]. However, despite several pathophysiological similarities existing between COVID-19 and non-COVID-19 ARDS, some substantial differences should be discussed (Figure 1 and Figure 2).

Gas-exchange is determined by ventilation and perfusion. Normal lung areas present a normal ventilation and perfusion (V/Q) ratio. In pathological conditions, major alterations may occur, explaining worsening in oxygenation: (1) increased lung regions with decreased ventilation to perfusion ratio from 0.1 to 0.001 (low V/Q ratio), and (2) non ventilated but perfused (“true shunt”) lung regions. The sum of the low V/Q and lung regions with “true shunt” determines the so called “venous admixture” [5,6,7]. The hypoxia due to low V/Q ratio may improve by administration of increased inspiratory oxygen fractions, while hypoxia due to “true shunt “does not improve by administration of increased inspiratory oxygen fractions. On the other side, lung regions with higher ventilation to perfusion ratio or ventilated but not perfused (“dead space”) increase the “wasted ventilation” leading to ineffective respiratory effort for carbon dioxide (CO_2_) washout. These lung regions do not substantially affect oxygenation [5,6,7].

### 2.1. Non-COVID-19 ARDS

Non-COVID-19 ARDS may be distinguished in “pulmonary” or “extrapulmonary” different phenotypes [8] that show distinctive histopathological features but are difficult to separate due to their overlap in clinical manifestations, being that most patients with pneumonia are also characterized by sepsis [8].

The “pulmonary” ARDS phenotype is characterized by a direct insult to the lung structure; it is mainly caused by pneumonia, leading to increased damage of the alveolar epithelium, prevalent neutrophils and fibrinous alveolar infiltrates, prevalent increases in compartmentalized inflammatory mediators, less interstitial and alveolar edema, normal capillary endothelium, and moderate increases in systemic inflammatory mediators. The more consolidated lung regions cause an increase of lung weight, while hypoxia is mainly due to variable distribution of ventilation and perfusion, depending to the pattern of lung injury (combination of low V/Q as well as “true shunt” lung areas). In this phenotype, higher positive end-expiratory pressure (PEEP) may lead to greater redistribution of ventilation and perfusion toward the non-dependent lung regions, with marked over-inflation of healthy lung regions. Prone position may improve oxygenation, mainly by partially redistributing pulmonary blood flow from dorsal to ventral lung regions and not by effective alveolar recruitment. This might be associated with decreased CO_2_ washout, since no increase in effective alveolar ventilation is expected [9,10].

On the contrary, the “extrapulmonary” ARDS phenotype is characterized by an indirect insult to the lung structure, and it is mainly caused by sepsis, leading to decreased damage of the alveolar epithelium, decreased neutrophils and fibrinous alveolar infiltrates, less compartmentalization of inflammatory mediators, higher interstitial and alveolar edema, altered capillary endothelium, and major increases in the systemic inflammatory mediators.

Increased lung weight is also consistent in “extrapulmonary” ARDS, but it is due to higher edema, leading to predominant alveolar collapse in the dependent lung regions in the supine position, associated with a gravitational distribution of perfusion; thus, hypoxia is prevalently due to true shunt. Higher PEEP may lead to effective alveolar recruitment, with greater redistribution of ventilation and perfusion toward the dependent lung regions, with less over-inflation of non-dependent lung regions. Prone position may improve oxygenation, mainly by recruitment of collapsed lung regions, while maintaining perfusion higher toward the dorsal lung regions. This might be associated with increased CO_2_ washout, since effective improvement in regional alveolar ventilation is expected [9,10].

Therefore, in traditional ARDS, higher regional perfusion and “true shunt” are mostly diverted to the dependent lung regions as edema is formed toward a ventral–dorsal gradient [11,12,13]. This causes the pressure applied on dependent lung regions to favor progressive atelectasis formation, while PEEP application keeps the dependent alveoli open, improving gas exchange and respiratory system compliance [11,12,13].

### 2.2. COVID-19 ARDS

COVID-19 ARDS may be considered as a typical example of “primary“ ARDS not associated with sepsis, at least in the initial stages. In COVID-19 ARDS, non-dependent aerated regions are mostly perfused, thus justifying a certain degree of hypoxic vasoconstriction in the dependent lung regions, resulting in a non-gravitational distribution of regional blood flow [13]. Unlike typical ARDS, in COVID-19, the lung compliance system does not deteriorate rapidly but gas-exchange impairment and hypoxia predominantly occur due to micro-thrombosis, dead space, and low V/Q lung regions [12,13,14,15].

In fact, COVID-19 ARDS is described by two different phenotypes (1 and 2) that represent an evolution of lung injury. Phenotype 1, typical of the early stages of the disease, is characterized by minor increases in lung weight, mainly due to poorly aerated (ground-glass radiological pattern) or consolidated lung regions, greater presence of aerated lung tissue, and minor changes in the respiratory and lung compliance, while hypoxia is mainly due to increased lung areas with low V/Q ratio due to lower ventilation in poorly aerated regions and/or increased perfusion in normally and poorly aerated lung regions. In this phase, beneficial response to higher oxygen fraction and effectiveness of non-invasive respiratory support is observed [14,15,16,17]. With the evolution of lung injury towards phenotype 2, the decrease in aerated lung regions and the increase in non-aerated but perfused ones lead to increased shunt, while the reduction in perfusion of non-aerated regions may limit the severity of hypoxia [17]. Phenotype 2 is characterized by a further reduction in aerated lung surface with a progressive reduction in the lung compliance, greater increase in the lung weight mainly due consolidated lung regions and non-perfused areas that become predominant. This causes the blood flow redistribution to these injured areas, leading to a worsening in oxygenation that can be caused by hypoxic vasoconstriction, thrombotic obstruction, or compression of capillaries [17]. Thus, the distribution of perfusion in COVID-19 ARDS follows a reverse gravitational pattern with a higher blood volume distributed in aerated (ventral) lung areas [13,15,17]. In this phase, no beneficial response to higher oxygen fraction and no effectiveness of non-invasive respiratory support is observed, while demanding prompt intubation if gas-exchange deteriorates and inspiratory effort increases [15,16,18]. The recruitment of COVID-19 ARDS (ranging from 1 to 6% of total lung weight) was lower than that reported in non-COVID-19 ARDS (ranging from 8% to 25% of lung weight [19,20]. Ball et al. showed that non-aerated tissue was reduced by increasing PEEP from 8 to 16 cmH_2_O, while normally aerated and hyper-aerated tissues increased both in recruiters and non-recruiters, and poorly aerated tissue decreased only in non-recruiters [20]. Moreover, alveolar recruitment was independent from the respiratory system compliance, and it slightly changed from 8 to 16 cmH_2_O of PEEP. Recruitment was prevalent in the dependent and caudal regions, but it did not correlate with the lung weight (which is associated with non-aerated tissue), changes in laboratory parameters, gas-exchange, and respiratory mechanics [20]. Higher PEEP decreased respiratory system compliance and improved oxygenation only at FiO_2_ 50% but not higher. Indeed, higher PEEP may lead to greater redistribution of ventilation and perfusion toward the non-dependent lung regions, with marked over-inflation of healthy lung regions. Prone position may improve oxygenation, mainly by partially redistributing pulmonary blood flow from dorsal to ventral lung regions, and not by effective alveolar recruitment. This might be associated with decreased CO_2_ washout, since no increase in effective alveolar ventilation is expected [15,16,17]. These findings support the hypothesis that the excess of lung weight in COVID-19 ARDS is not due to edema, and that PEEP might improve oxygenation by acting more on V/Q matching in areas with low V/Q than on alveolar recruitment [20].

## 3. The Pathophysiology of Severe Pneumonia/ARDS Due to COVID-19: Correlation from Lung CT Scan and Histology Findings with Clinical Phenotypes

Severe COVID-19 pneumonia shares several histological and imaging features with non-COVID-19 ARDS, since both are characterized by inflammatory lung injury, progressive parenchymal stiffening and consolidation, alveolar and airway collapse, altered vascular permeability, and a pathological picture of diffuse alveolar damage (DAD) [4]. Usually, three histopathological phases are described in traditional ARDS: (1) acute early exudative (1–7 days) with intra-alveolar edema, interstitial widening, formation of hyaline membranes, epithelial disruption, and alveolar distortion; (2) subacute proliferative phase (1–3 weeks) with myofibroblast proliferation and hyperplasia of the epithelial lining; and (3) late fibrotic phase (>3 weeks) characterized by collagen deposition, progressive obliteration of alveolar spaces, and interstitial fibrosis [4]. Associated thrombi can result as expression of an altered coagulation cascade, which is common in COVID-19 [3] (Table 1).

In the monitoring of severe COVID-19, imaging techniques play a pivotal role. Chest X-ray and computed tomography (CT) scan are useful for diagnosis and clinical management, usually showing coexistence of typical aspects of viral pneumonia (i.e., ground glass opacities with or without consolidations and crazy-paving opacities with a subpleural distribution) and ARDS features (Figure 3). From a clinical point of view, distinct chest CT patterns have been categorized into different COVID-19 phenotypes that can benefit from different clinical management: phenotype 1 (or type L, characterized by low elastance, high lung compliance, low ventilation/perfusion ratio, low weight, low recruitability) [21,22], typical of mild to moderate disease that can initially benefit from noninvasive respiratory support, presents as multiple, often bilateral, focal over-perfused ground glass opacities and normally aerated zones; phenotype 2 (or type H, i.e., high elastance, low lung compliance, high ventilation/perfusion ratio, high lung weight, high recruitability) [21,22], more typical of the most severe COVID-19, shows non-homogeneously distributed hyper- and hypo-perfused areas with predominant atelectasis and a patchy ARDS-like pattern [16,22,23,24] Several studies reported that phenotype L is most frequent (70–80% of patients) than phenotype H (20–30%) [12,21,22,23,24]. In their study, Gattinoni et al. reported that the transition from phenotype L to type H may be due to both the evolution of COVID-19 pneumonia and the injury attributable to high-stress ventilation. The same group concluded that type L and H are best identified by CT scan, which could be useful also to recognize other possible comorbidities [12]. In line with recent assumptions, the proposed COVID-19-related phenotypes may be considered as the two extremes of a unique evolving disease [21,22,23,24]. A recent article demonstrated that phenotype H was more common among females and was associated with a higher risk of ARDS and burden of comorbidities, including diabetes. Furthermore, patients classified as phenotype H were liberated from mechanical ventilation later and characterized by higher mortality as compared to phenotype L patients [12,21,22,23,24].

Finally, phenotype 3 (fibroproliferative) includes the low-specificity CT findings of an advanced COVID-19 pneumonia, where a dysregulated fibrotic response follows the extensive damage to lung scaffold and vasculature [23,24].

The necessity to develop radiological tools which can predict which patients are at higher risk for development of a rapid and fatal progression of the disease during the acute phase quickly emerged. Severity scores based on chest X-ray [25,26] or CT scan have been proposed to stratify the patients by quantifying the extension of the disease and to predict both the need for ICU referral and death at the time of hospital admission [27,28]. The radiological long-term findings are represented by interlobular septal thickening, reticular opacities, and subpleural curvilinear opacities in association with bronchial dilatation and distortion (Figure 3, Table 1). Reticulations (fibrotic-like changes) can either be an early sign of post-ARDS fibrosis, as described in previous coronavirus outbreaks (SARS and Middle East respiratory syndrome, MERS), or they can be related to the evolution of the organizing pneumonia [29]. In SARS-CoV-2 infection complicated by ARDS [30], phenotype 3 is found in 35% [31] to 72% [32] of patients at six months follow-up. On the contrary, the incidence of pulmonary fibrosis was about 4% during previous coronavirus-related epidemics [33], while it reached 25% [34] after H1N1 pneumonia. Several factors have been correlated with the risk of developing fibrosis and persistent functional deficit in COVID-19, i.e., the severity of infection, the need for ICU admittance and mechanical ventilation, older age, higher body mass index, and smoking history [29,35]. Hence, patients who recover from SARS-CoV-2 related ARDS but experience persistent respiratory symptoms need to undergo a radiological follow-up to determine whether alterations that can anticipate the establishment of fibrosis exist [29]. Despite high resolution CT (HRCT) being considered the gold standard for diagnosis, staging, and follow-up of COVID-19 [36,37,38], it requires patient transportation and radiation exposure. Thanks to the elevated sensitivity and specificity of ultrasound in the field of pleural, parenchymal, and interstitial abnormalities, the whole international literature considers lung ultrasound (LUS) a useful support for the diagnosis of COVID-19 pneumonia in the emergency setting as well as for the clinical and therapeutic follow-up [39,40,41,42,43,44,45]. Indeed, LUS can be easily used at a patient’s bed without X-ray exposure, and it can be repeated several times, for example in pregnant women and children [46]. In patients with COVID-19 pneumonia, also during the preclinical period, a specific LUS pattern is represented by pleural thickening, pleural interruptions and effusions, multifocal B lines, and small opacities [46,47,48,49]. To date, there is no single established algorithm for performing LUS in patients with COVID-19 disease, despite several approaches having been proposed, such as scanning 12-, 14-, or 16-lung zones [46,47,48,49]. Several studies showed that in COVID-19 pneumonia still at an early stage, LUS can detect early lung lesions even when CT scan is almost normal. Furthermore, there was a relationship between the abnormalities detected by the LUS and CT, especially between the presence of more than 2 B-lines on LUS and ground glass opacity areas on CT and between subpleural consolidations on LUS and consolidation areas on CT [36,37,38,50]. Recently, Lupia et al. enrolled 208 patients with COVID-19 respiratory symptoms who underwent nasal swab and LUS according to a scanning protocol [47]. In 20% of patients who had a false negative nasal swab, ultrasound showed typical parenchymal abnormalities for COVID-19. Due to the persistence of pathological ultrasound aspects after the acute phase, thoracic ultrasonography has also been chosen among the tools for COVID-19 follow-up after patient discharge [47,51]. Thus, when combined with clinical data, LUS can provide a potent monitoring aid in patients with suspected COVID-19 pneumonia, reflecting and anticipating CT findings. However, LUS does not add much value in determining invasive versus non-invasive management in these patients, and it is extremely challenging to use as a monitoring tool. LUS is operator-dependent, and it should be performed and interpreted by an experienced professional. Furthermore, LUS only evaluates pleural and subpleural changes, while deeper lesions could be missed since ultrasound waves cannot penetrate highly aerated lungs. Ultrasound findings also depend on the clinical characteristics of the patient; obesity and subcutaneous emphysema can interfere with passage of US waves and make the assessment more difficult.

## 4. Noninvasive Measurement of Compliance: Role of the Forced Oscillation Technique (FOT)

The measurement of compliance has been the object of intense study because normal or close-to-normal compliance has been hypothesized to benefit different respiratory supports in COVID-19 ARDS [22]. At the present time, compliance can be noninvasively measured by the Forced Oscillation Technique (FOT) parameter reactance. FOT is a non-invasive, non-expensive, and fast approach which measures lung function using sinusoidal pressure waves generated by a loudspeaker and forced into the lung during tidal breathing, requiring no effort maneuvers [52]. Furthermore, FOT is a simple approach to measure lung mechanics, since it needs the patient to have spontaneous tidal volume breathing for a maximum of 10 breath, and it uses a portable instrument. We showed in our preliminary abstract that it is feasible bedside in patients with critical COVID19 pneumonia [22]. A pressure–flow transducer measures inspiratory and expiratory flow and pressure, which are then separated from the breathing pattern by signal filtering. The fraction of pressure over flow at a specific frequency is called impedance (Z) and is measured in cmH_2_O⋅L^−1^⋅s^−1^ or kPa⋅L^−1^⋅s^−1^ [53]. The impedance of the respiratory system (Zrs) is the sum of the respiratory resistance (Rrs) and the respiratory reactance (Xrs) measured over a range of increasing frequencies (i.e., from 3 to 35 Hz) [52]. Reactance (Xrs) is the part of the impedance of the respiratory system (Zrs) related to the elastic properties of the lung periphery, and it is the sum of inertance (I, the force of the accelerating air column) and capacitance (C, related to the elastic properties of lung periphery). At low frequencies, the capacitive properties of the small peripheral airway predominate; on the contrary, inertance increases its contribution to reactance at higher frequencies. Therefore, reactance at 5 Hz (X_5_) provides important information about distal airways. By convention, capacitance is defined negative, and inertance positive, and hence their sum is negative at low frequencies and positive at high frequencies. Conditions that reduce the elasticity of the lung, such as fibrosis and hyperinflation, make the capacitance increasingly negative, i.e., more negative *X*_5_ values [52]. FOT has been applied to several clinical settings, as well as in lung function laboratories, with special regard to airway diseases and pediatric respiratory diseases [54,55], but it has also been used during CPAP and NIV [56,57,58,59]. ARDS has been investigated by FOT in animal models and in the newborn. In an experimental model of lung recruitment/derecruitment, C_x5_ (oscillatory compliance derived from X_5_) resulted in being highly specific in detecting the loss of ventilated lung [60] (Figure 4). Reactance has been suggested to have advantages over dynamic compliance (C_dyn_), allowing for the conduction of studies to assess intra-tidal events (recruitments or over-distension) while maintaining the breathing pattern [60]. FOT has also been shown to allow the optimizing of PEEP, both in preterm infants, where surfactant deficiency is associated with increased probability of alveolar instability, atelectasis, and lung volume derecruitment [61], and in the animal model [62]. The feasibility of FOT in severe COVID-19 pneumonia needing noninvasive support was recently studied at the Pulmonology Department of the University of Trieste (unpublished data), showing that it can provide useful additional pathophysiological parameters in patients with ARF undergoing high-flow nasal cannula (HFNC) oxygen support. Further studies are required to assess feasibility and reproducibility of this technique in the management of patients with ARF with impeding necessity of NIRS.

## 5. Noninvasive Respiratory Support in Severe COVID-19

NIRS represents a possible first step in the management of COVID-19 patients [63]. Several observations reported that patients with COVID-19 phenotype 2 (or H), who are at higher risk of disease progression, could be in need of early intubation, while those who present with phenotype 1 (or L) can be initially managed with NIRS [12,23,24]. Indeed, COVID-19 patients receiving invasive mechanical ventilation showed a decrease in pulmonary volume and an increase in regions of poorly aerated or non-aerated lung tissue compared to those receiving NIRS [17]. Therefore, the invasive approach should be reserved to severe cases. The recognition of patients who are at higher risk of NIRS failure is challenging [63], and the possible progression to patient self-inflicted lung injury (P-SILI) should take into account. This can be caused by several mechanisms including increased lung stress and strain, inhomogeneous distribution of ventilation, changes in lung perfusion, or patient–ventilator asynchronies [64]. Therefore, a strategy ranging from low to high-flow oxygen support, CPAP, or NIV and taking onto account invasive mechanical ventilation only in cases of failure of the previous NIRS strategies or phenotypes 2 or 3 is advisable (Figure 5) [18]. Before considering intubation in patients who are at risk of NIRS failure, a trial period of awake prone positioning can be also carried out without delaying intubation if the oxygenation progressively worsens [18]. When providing invasive mechanical ventilation in COVID-19, low tidal volumes (Vt) (4–6 mL/kg of predicted body weight [PBW]) with low plateau pressure (<28–30 cmH_2_O) and moderate levels of PEEP (10 to 15 cmH_2_O) should be applied [18]. In COVID-19 pneumonia with preserved lung compliance and few atelectasis areas, it is suggested to use low to moderate PEEP levels. On the contrary, in cases of low compliance phenotypes with extensive atelectasis, a strategy comprising moderate PEEP levels, prone positioning, and occasional recruiting maneuvers might be indicated [18]. Respiratory physiotherapy has also a key role in the management of the disease, yielding a potential for gas-exchange improvement [65,66]. In summary, COVID-19 pneumonia is characterized by progressive increases in lung weight due to inflammatory edema affecting gas-exchange. In early phases, low V/Q areas are predominant with good response to higher oxygen fraction and moderate levels of PEEP, while in the later stages, true shunt is predominant with poor response to higher oxygen fractions and PEEP, not exceeding 15 cmH_2_O. The worsening of oxygenation during noninvasive respiratory support requires early intubation and initiation of invasive mechanical ventilation.

## 6. Physiological Parameters Monitoring during Noninvasive Support for Acute Respiratory Failure Due to Severe COVID-19

Preserving spontaneous breathing in patients with ARF under non-invasive respiratory assistance may represent a risky gamble when hypoxemia is severe. On the one hand, several positive physiological effects have been described including the avoidance of deep sedation and/or myorelaxants drugs, the prevention of muscle mass loss, the sparing of diaphragmic function, and the reduction of delirium onset [67,68]. However, a growing body of evidence derived from animal models and clinical investigations on classical ARDS has strengthened the hypothesis that the presence of intense respiratory effort producing excessive negative pleural pressure swings (P_pl_) plays a critical role in the onset and progression of lung and diaphragm damage, especially when lung impairment is severe [69,70,71,72]. This unfavorable mechanical condition predisposes patients to the so-called P-SILI, whose underlying mechanisms differentiate from those sustaining the well-known model of ventilatory induced lung injury (VILI). When lungs are healthy, the transpulmonary pressure (P_L_) generated by diaphragmatic contraction is uniformly distributed over the entire lung surface. The elastic response of the lung follows a “liquid-like” behavior without local alveolar over-distention [73,74]. When acute lung injury occurs, local inflammation and alveolar edema make lung tissue inhomogeneous. The transmission of the forces applied to pulmonary parenchyma during spontaneous breathing becomes asymmetrical, and the elastic response of the lung follows a “solid-like” behavior. In particular, the negative swing in pleural pressure generated during active inspiration is not uniformly distributed, being magnified in dependent regions and alongside the diaphragmatic interface, where high values of P_L_ are concentrated [75]. The unbalanced application of physical forces during spontaneous breathing causes a pressure gradient between nondependent and dependent lung zones, resulting in a disproportionate distribution of tidal volume with an unsafe local stretch of the dependent lung (pendelluft phenomenon) [75]. A significant drop in intrathoracic pressure due to intense inspiratory effort can result in negative changes in alveolar pressure. The following increase in transmural vascular pressures in pulmonary capillaries predisposes one to the development of pulmonary edema [76]. Disparate radial traction forces, applied in the corner vessels adjacent to stress raisers, may generate a siphoning effect of blood towards areas of higher P_L_ (pendelblut phenomenon) [77]. Load-induced diaphragm injury can occur during intense spontaneous breathing, as suggested by the radiological signs of muscle edema [72] and the histological evidence of fiber disruption, sarcomeric disturbance, and amplified inflammation [78]. These biophysical insults related to the magnitude of inspiratory effort and to the following lung parenchymal stretch, alveolar edema, and diaphragmatic overload can result in a pattern of progression from the initial lung damage, worsening ventilatory and clinical outcomes [70]. The characteristics of respiratory drive activation along with the mechanical and clinical consequence of spontaneous breathing in COVID-19 patients have become matters of investigation, giving the peculiar pathophysiological features of this unforeseen form of ARDS [79]. Since the onset of the pandemic, several clinical observations have pointed out that a considerable number of patients experiencing COVID-19 pneumonia in the early phase do not present subjective dyspnea despite severe hypoxemia, this condition being defined as “happy hypoxia” [80,81,82]. Although the mechanisms beyond the limited shortness of breath are not fully understood, it has been hypothesized that damage of the C-pulmonary afferent fibers driven by the inflammatory cascade or by direct viral involvement may affect the coupling between bio-mechanical stimuli and respiratory drive activation [83]. A recently published matched study comparing COVID-19 receiving non-invasive respiratory support with moderate to severe ARF with classical ARDS [84] showed a relatively low activation of respiratory drive in COVID-19 patients during the early phase (median change in esophageal pressure [ΔP_es_] 12.5 cmH_2_O, respiratory rate 28 bpm, Vte 9.2 mL/kg of PBW versus ΔP_es_ 32 cmH_2_O, RR 35 bpm, expiratory tidal volume (Vte) 10.9 mL/kg of PBW), which is in line with the aforementioned concept of “happy hypoxia” [82] and underlines the mismatch between central drive activation and moderate to severe hypoxia, at variance with the typical form of ARDS. At least theoretically, these data seem to suggest that in the very early phase of assisted spontaneous breathing, the role of SILI in determining further lung damage is not as crucial in early COVID-19 ARDS as in typical forms of ARDS. In agreement with this, the dynamic compliance of the respiratory system was twice as high in COVID-19 patients as in those with traditional ARDS (55 vs. 25 mL/cmH_2_O) at comparable arterial partial pressures of oxygen/fraction of inspired oxygen (PaO_2_/FiO_2_) ratios in the early stage of the disease. However, even if respiratory drive resulted in less stress as compared with ARDS, the application of NIV in these patients determined a significant reduction of inspiratory effort with values of ΔP_es_ close to physiological ranges (6–10 cmH_2_O). Moreover, unpublished data on lung behavior during spontaneous breathing, according to different PEEP levels in this cohort, showed that elevated values of PEEP were inversely correlated with the relative change of the PaO_2_/FiO_2_ ratio and dynamic lung compliance after NIV (Figure 6). In contrast, typical ARDS patients showed a favorable association between PEEP values and gas exchange 2 h after starting NIV, suggesting a different behavior in response to recruitment. These findings are in line with those reported by Coppola et al., who showed that in 23 critically ill patients with COVID-19 pneumonia, increasing PEEP from a low (5 cmH_2_O) to a higher (15 cmH_2_O) level led to a significant deterioration in lung mechanics as assessed via esophageal manometry [85]. The concept of mechanical power has recently been developed to explore the interaction between ventilatory support and lung damage. In particular, the degree of VILI has been related to the amount of energy transferred from the mechanical ventilator to the respiratory system [86]. Assuming that the amount of energy to which the lung is subjected, even during assisted spontaneous breathing, may be crucial in P-SILI development, a simple surrogate of mechanical power can be derived by replacing the change in airway pressure during inspiration with dynamic transpulmonary pressure [87]. In patients with COVID-19, the baseline value of dynamic mechanical power was considerably lower than in ARDS (27 vs. 95 J/min, *p* < 0.0001). After a 2 h NIV trial, the dynamic mechanical power showed a significant increase in COVID-19 patients alone. This may suggest an unfavorable interaction between potential and kinetic energy transferred from the respiratory muscles and the mechanical ventilator to the lungs of these patients, at least in the early phase of the disease when respiratory drive is still preserved. In more advanced stages of COVID-19 pneumonia, following the phenotype transition with increase in lung weight and relative drop in lung compliance, an increase in respiratory drive has been documented and correlated with worsening of respiratory function during attempts to wean patients from mechanical ventilation [88]. A recent computational study has shown how in a model correlated to COVID-19 patients, when intense inspiratory effort (namely pleural pressure swing) was reached, the physical forces produced were comparable with those associated with VILI during mechanical ventilation [89]. The authors concluded that inspiratory effort in these patients should be carefully monitored and controlled to reduce the risk of lung injury. Given that esophageal pressure swings mirror P_pl_ during spontaneous breathing, esophageal manometry by means of an esophageal balloon catheter is considered a reliable method to quantify the magnitude of inspiratory effort [90]. In typical ARDS, an intense inspiratory effort, as documented by high ΔP_es_ swings, is associated with unfavorable ventilatory outcomes [70]. A pressure threshold to be considered harmful and predisposing to the onset of P-SILI in COVID-19 patients is still to be defined. However, the continuous monitoring of inspiratory effort through esophageal manometry may allow the impact of non-invasive respiratory support to be assessed and to improve the ventilatory management of spontaneously breathing patients with severe COVID-19. In particular, (1) the quantification of inspiratory effort may allow for a precise characterization of respiratory drive, whose hyperactivation require immediate intervention in order to reduce both lung tissue stress and the increase in pulmonary trans-vascular pressures sustained by vigorous breathing effort [89,91]; (2) the assessment of changes in ΔP_es_ following the application of non-invasive respiratory support may help in early discriminating between good and low responders to respiratory assistance, avoiding the use of positive pressure when unneeded and intercepting non-responders to NIRS who may benefit of an upgrade to invasive mechanical ventilation (Figure 7); (3) a continuous monitoring of respiratory effort may inform the changes in the respiratory mechanics of the patient; in particular, an abrupt increase in ΔP_es_ may suggest a rapid disturbance of respiratory system compliance, mirroring the transition from a lung with fluid-like behavior to parenchymal solid-like elastic properties; (4) esophageal manometry could be useful to obtain information on lung mechanical features and the relative response to positive pressure application on the respiratory system. Indeed, a surrogate of lung compliance, namely dynamic compliance, can be derived comparing the values of P_L_ with Vte. Furthermore, a simplified surrogate of mechanical power, defined as dynamic mechanical power, can be calculated as 0.098 * RR * Vte * (ΔP_L_ + PEEP) [87]. Although approximate, this index may represent a reliable estimate of the amount of energy transferred from respiratory muscle and ventilatory assistance to the lung during assisted spontaneous breathing. All these physiological variables may inform the clinician of lung recruitability at bedside, allowing for the optimization of PEEP during non-invasive respiratory assistance, while unfavorable changes in dynamic compliance and mechanical power may suggest limited lung recruitability and forecast the risk of local overdistension. Despite systematic assessment of inspiratory effort by means of esophageal monitoring being certainly appealing, this technique seems not easy to implement in everyday clinical practice [92]. This is partially due to technical issues such as the correct insertion and proper placement of the probe, which influences the accuracy and interpretation of measurements. Moreover, the procedure itself could be difficult when respiratory drive is markedly activated, as it may happen in awake patients with respiratory distress and severe gas exchange impairment [93]. Finally, the maneuver is invasive, and it may cause discomfort or potential side effects in awake patients with ARF [94].

## 7. Summary and New Perspectives

As we have established, severe COVID-19 presents different pathophysiological phases that partially resemble typical ARDS and have been categorized into three radiological patterns and clinical phenotypes (Table 1). During the mildest forms or earlier phases of COVID-19 ARDS, the compliance of the respiratory system is not affected, but gas-exchange impairment mostly occurs due to an altered V/Q ratio. On the contrary, with the evolution of lung injury towards most severe or advanced forms, the decrease in aerated lung regions and the increase in non-aerated ones due to inflammatory edema lead to increased shunt. The clinical response to the application of external PEEP during mechanical ventilation depends on lung recruitability and compliance. Nevertheless, a growing body of evidence suggests that intense respiratory effort producing excessive negative pleural pressure swings (P_pl_) plays a critical role in the onset and progression of lung and diaphragm damage in patients treated with NIV. The routine use of respiratory monitoring would be a powerful means to avoid the useless continuation of NIRS in patients who are at higher risk for respiratory deterioration and could benefit from early initiation of invasive mechanical ventilation. Forced Oscillation Techniques (FOTs) could help in identifying the subset of high-compliance patients who can benefit from NIRS continuation. Esophageal monitoring is an appealing technique for the systematic assessment of inspiratory effort, despite it implying technical issues that are difficult to overcome in everyday clinical practice. Advanced respiratory monitoring, including novel noninvasive inspiratory effort devices, could be useful to monitor patients with ARDS at risk for an injurious (spontaneous or assisted) ventilation, as may happen in the COVID-19 pandemic when the use of NIRS is extensive [6]. A more feasible method for measuring inspiratory effort without the technical issues of esophageal manometry could ameliorate P-SILI monitoring and help predict respiratory deterioration in the clinical setting. Further research is needed on non-invasive methods to quantify the magnitude of inspiratory effort to optimize the ventilatory management of patients with ARDS of different etiology, including COVID-19.

## 8. Conclusions

COVID-19 ARDS is a new respiratory disease that arguably differs from traditional ARDS. For this reason, a personalized respiratory and monitoring strategy in patients with SARS-CoV-2 infection is paramount. An individualized ventilatory approach based on lung physiology, morphology, imaging, and identification of biological phenotypes may improve COVID-19 outcomes while individualizing mechanical ventilation practices.

## Figures and Tables

**Figure 1 jcm-11-01704-f001:**
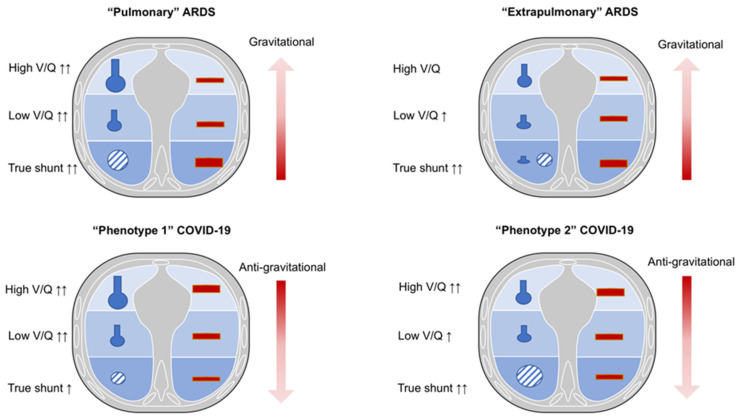
Ventilation/perfusion characteristics in non-COVID-19 and COVID-19 ARDS in supine position. Legend. Pulmonary ARDS: in supine position, low ventilation/perfusion ratio (V/Q) is prevalent. Higher regional perfusion and true shunt are mostly diverted in the dependent lung regions. Extrapulmonary ARDS: in supine position, true shunt is prevalent with possible alveolar collapse. The pulmonary blood flow and shunt mainly distribute toward the dependent lung regions. COVID-19 phenotype 1: in supine position, a low V/Q is prevalent with anti-gravitational distribution of pulmonary blood flow. COVID-19 phenotype 2: in supine position, true shunt is increased, while the pulmonary blood flow is anti-gravitational. V/Q, ventilation/perfusion; COVID-19, coronavirus disease 2019; ARDS, acute respiratory distress syndrome.

**Figure 2 jcm-11-01704-f002:**
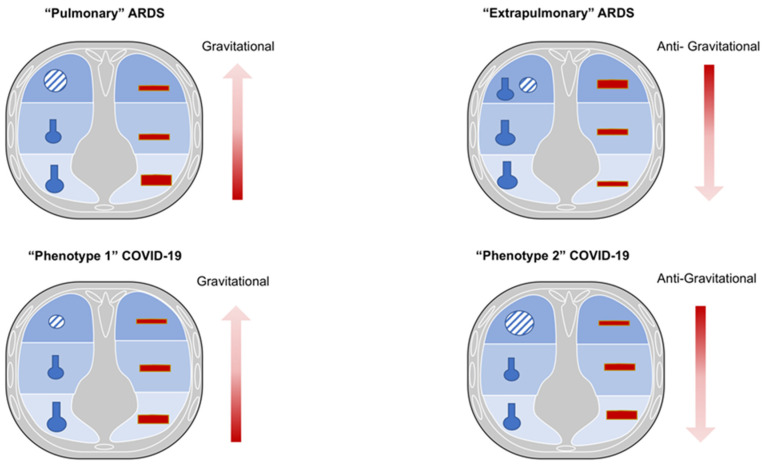
Ventilation/perfusion characteristics in non-COVID-19 and COVID-19 ARDS in prone position. Legend. Pulmonary ARDS: in prone position, oxygenation may improve due to partial redistribution of pulmonary blood flow toward ventral regions, and not by effective alveolar recruitment. This might be associated with decreased carbon dioxide washout. Extrapulmonary ARDS: in prone position, oxygenation can improve because of possible alveolar recruitment of collapsed alveoli, while maintaining perfusion higher toward the dorsal lung regions. COVID-19 phenotype 1: in prone position, oxygenation may improve, partially redistributing pulmonary blood flow that remains anti-gravitational. COVID-19 phenotype 2: in prone position, oxygenation may improve, partially redistributing pulmonary blood flow from dorsal to ventral lung regions, and not by effective alveolar recruitment. This might be associated with decreased carbon dioxide washout. V/Q, ventilation/perfusion; COVID-19, coronavirus disease 201; ARDS, acute respiratory distress syndrome.

**Figure 3 jcm-11-01704-f003:**
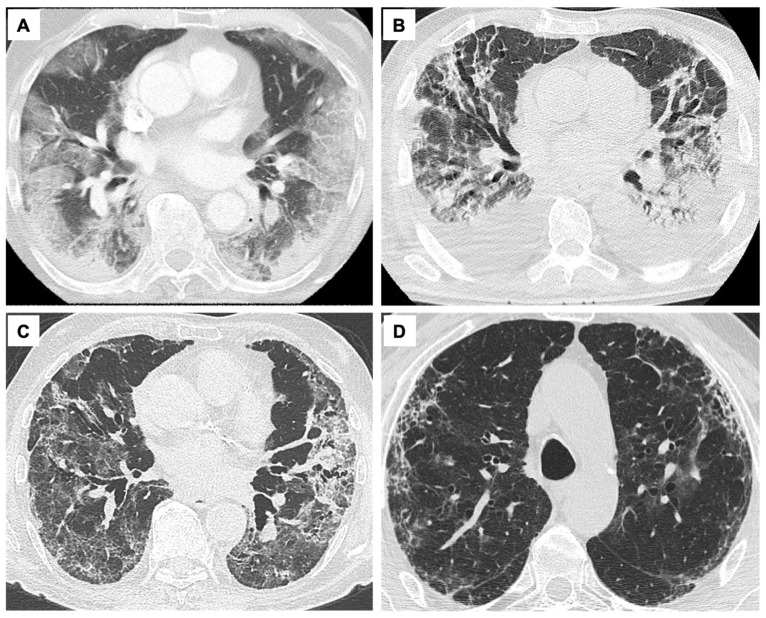
COVID-19 radiological phenotypes at computed tomography and follow-up. Legend. Axial chest HRCT showing the main features of COVID-19 phenotypes. (**A**) Phenotype 1: diffuse bilateral ground glass opacities and consolidation. (**B**) Phenotype 2: patchy, dependent, bilateral areas of parenchymal consolidations. (**C**) Phenotype 3: diffuse interlobular septal thickening, bronchial dilatation/distortion, and perilobular fibrosis. (**D**) Persistence of thin subpleural reticular and curvilinear opacities after 9 months from ARDS resolution. HRCT, high resolution computed tomography scan; COVID-19, coronavirus disease 2019; ARDS, acute respiratory distress syndrome.

**Figure 4 jcm-11-01704-f004:**
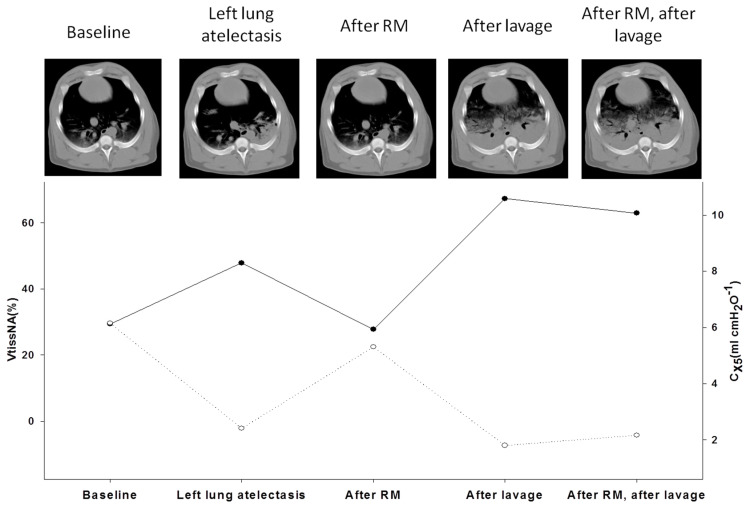
Correlation between oscillatory compliance and lung aeration. Legend. Upper panel. CT scans of an animal model of induced lung injury; from left to right: baseline, during left lung atelectasis induced by 10 min of single-lung ventilation at 100% oxygen, after a RM, post broncho-alveolar lavage and after RM after broncho-alveolar lavage. Lower panel: corresponding VtissNA% (closed symbols, solid line) and C_x5_ (open symbols, dotted line) showing that oscillatory compliance and lung derecruitment show a specular trend being inversely correlated. (With permission, courtesy of Prof. Dellaca [60]). C_x5_, 5Hz oscillatory compliance; VtissNA%, percentage of non-aerated tissue volume; CT, computed tomography; RM recruitment maneuvers.

**Figure 5 jcm-11-01704-f005:**
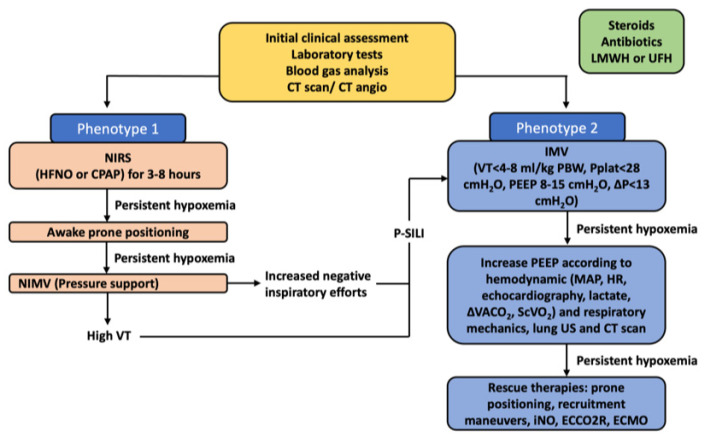
Suggested algorithm for the ventilatory management of COVID-19 ARDS. Legend. Algorithm for the ventilatory management of COVID-19 ARDS. NIRS, non-invasive respiratory support; HFNO, high-flow nasal oxygen; CPAP, continuous positive airway pressure; NIMV, non-invasive mechanical ventilation; IMV, invasive mechanical ventilation; V_T_, tidal volume; PBW, predicted body weight; PEEP, positive end-expiratory pressure; P, driving pressure; p-SILI, patient self-inflicted lung injury (patients on NIPPV who are generating massive negative pressures and high tidal volumes in the range of 10–12 cc/kg IBW); MAP, mean arterial pressure; HR, heart rate; ΔV_A_CO_2_, venous–arterial carbon dioxide tension difference; ScVO_2_, central venous oxygen saturation; US, ultrasound; CT, computed tomography; iNO, inhaled nitric oxide; ECCO2R, extracorporeal carbon dioxide removal; ECMO, extracorporeal membrane oxygenation; LMWH, low-molecular weight heparin; UFH, unfractioned heparin.

**Figure 6 jcm-11-01704-f006:**
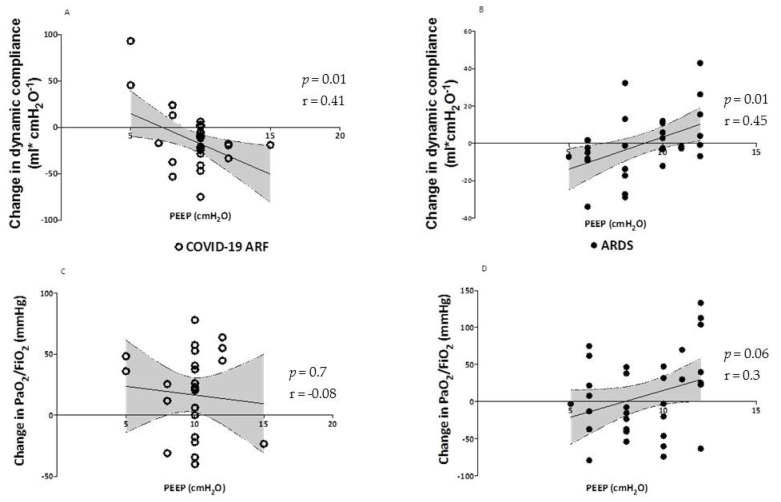
Correlation between PEEP and dynamic compliance and PaO_2_/FiO_2_ ratio in COVID-19 and non-COVID-19 ARDS. Legend. Correlation between PEEP values and change in both dynamic compliance and PaO_2_/FiO_2_ ratio in COVID-19 pneumonia (panel **A** and **C**, respectively) and ARDS (panel **B** and **D**, respectively) patients. PEEP, positive end-expiratory pressure; COVID-19, coronavirus disease 2019; ARDS, acute respiratory distress syndrome.

**Figure 7 jcm-11-01704-f007:**
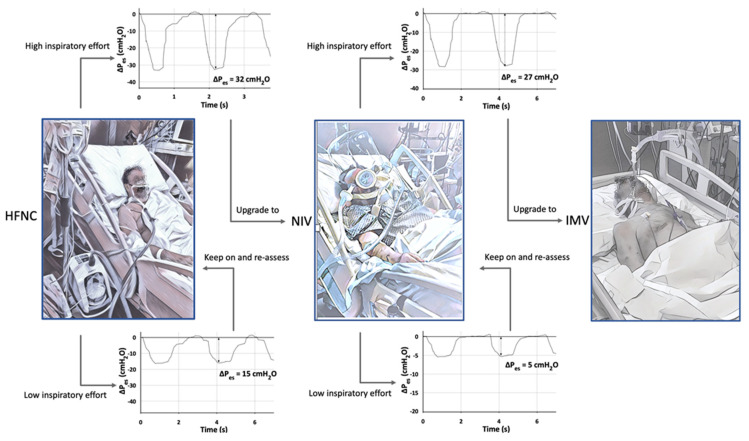
Practical flow-chart of respiratory assistance for patients with COVID-19 acute respiratory failure based on inspiratory effort assessment. Legend. HFNC trial might be started with close monitoring of inspiratory effort. If low values of ΔP_es_ are detected, HFNC should be kept with close monitoring of esophageal pressure swings and gas exchange. In case of high inspiratory effort, non-invasive respiratory assistance should be upgraded to NIV. If ΔP_es_ is reduced by positive pressure application, NIV might be continued with continuous monitoring of inspiratory effort. In case NIV fails to reduce inspiratory effort, a rapid switch to IMV should be considered. HFNO, high-flow nasal oxygen; NIV, non-invasive mechanical ventilation; MV, invasive mechanical ventilation; ΔP_es_, esophageal pressure.

**Table 1 jcm-11-01704-t001:** Characteristics of non-COVID-19 and COVID-19 ARDS.

Disease Type		Histopathological Findings	Radiological Findings	Clinical Characteristics and Management
** *Non-COVID-ARDS (DAD)* **	**“Pulmonary” ARDS**	Alveolar epithelial damage and collapse Neutrophilic infiltrates in terminal bronchioles and surrounding alveoli with confluence of infiltrates between adjacent lobules (prevalent) Fibrinous exudates in alveoli Interstitial edema (rare) Increased collage in the interstitial space Altered type I and II pneumocytes with nuclear atypia and hyperplasia (necrosis type I cells, proliferation type II cells) Apoptotic neutrophils (prevalent) Normal capillary endothelium Mallory like inclusions in type II pneumocytes Proliferation of fibroblast and myofibroblast Possible DAD (hyaline membrane plus intra-alveolar edema, necrosis of alveolar type I cells, proliferation of alveolar type II cells, interstitial proliferation of fibroblast and myofibroblast, interstitial fibrosis)	Asymmetric consolidation and ground-glass opacification Possible pleural effusion and emphysema Predominance of air bronchograms and pneumomediastinum	**Clinical manifestation** Hypoxia due to low V/Q and true shunt lung areas **Suggested treatment** IMV (low V_T_ 4–6 mL/kg PBW, higher PEEP, low Pplat < 28–30 cmH_2_O, ΔP < 13 cmH_2_O) Prone position (to redistribute pulmonary blood flow from dorsal to ventral lung regions, decreased CO_2_ washout, no improvement in regional alveolar ventilation)
	**“Extrapulmonary” ARDS**	Alveolar epithelial damage and collapse Neutrophilic infiltrates (rare) Fibrinous exudates in alveoli (rare) Interstitial edema (prevalent) Increased collage in the interstitial space Normal type I and II pneumocytes Apoptotic neutrophils (rare) Damaged capillary endothelium Possible DAD (hyaline membrane plus intra-alveolar edema, necrosis of alveolar type I cells, proliferation of alveolar type II cells, interstitial proliferation of fibroblast and myofibroblast, interstitial fibrosis)	Symmetric ground-glass consolidation (mainly distributed in the middle-basal levels and vertebral position) and opacification (greater in the central third of the lung than in the sternal or vertebral third without significant craniocaudal predominance) Possible pleural effusion and emphysema	**Clinical manifestation** Hypoxia due to alveolar collapse of dependent lung regions with gravitational distribution of perfusion and true shunt lung areas **Suggested treatment** IMV (low V_T_ 4–6 mL/kg PBW, higher PEEP, low Pplat < 28–30 cmH_2_O, ΔP < 13 cmH_2_O) Prone position (recruitment of collapsed areas maintaining higher perfusion toward dorsal lung regions, increased CO_2_ washout, improvement in regional alveolar ventilation)
** *COVID-19 ARDS* **	**Phenotype 1**	Alveolar collapse and rupture Intra-alveolar hemorrhage Hyaline tissue formation (rare) Microthrombi, vasculitis or vascular thrombosis Polymorphonuclear and monocytes infiltration (initial) SARS-CoV-2 replication in type II pneumocytes Reactive pneumocytes with nuclear atypia and hyperplasia Mallory like intracytoplasmic inclusions in type II pneumocytes Masson’s bodies	Multiple focal over perfused ground glass opacities and normally aerated areas Possible diversion of ventilation toward non-dependent aerated lung regions and reduction in pulmonary perfusion due to increased airway pressure Collapse of capillaries and/or micro-thrombosis and formation of no recruitable atelectasis	**Clinical manifestation** Normal compliance of the respiratory system Hypoxia (increased areas with altered V/Q ratio) **Suggested treatment** NIRS (HFNC, CPAP, NIV) with high FiO_2_ and respiratory monitoring (i.e., clinical deterioration, gas exchange, FOT, esophageal manometry) IMV when NIRS failed (using lower PEEP)
	**Phenotype 2**	Alveolar collapse and rupture Intra-alveolar hemorrhage Hyaline tissue formation (prevalent) Microthrombi, vasculitis or vascular thrombosis Early fibroblastic interstitial fibrosis, septal and para-septal reparative fibrosis Polymorphonuclear and monocytes infiltration (prevalent) Reactive pneumocytes with nuclear atypia and hyperplasia Mallory like intracytoplasmic inclusions in type II pneumocytes Masson’s bodies	Patchy ARDS-like pattern Inhomogeneously distributed and hyper/hypo-perfused areas Increased lung weight and consolidated Non-aerated lung regions (dependent lung regions)	**Clinical manifestations** Decrease of aerated lung regions Impairment of compliance of the respiratory system Increased shunt (blood flow redistribution to injured areas with hypoxic vasoconstriction, thrombosis, and compression of capillaries) **Suggested treatment** IMV (low V_T_ 4–6 mL/kg PBW, higher PEEP to redistribute ventilation and perfusion, Prone positioning (partial redistribution from dorsal to ventral areas, no effective recruitment)
	**Phenotype 3/F**	Hyaline membranes Fibroblastic interstitial fibrosis, septal and para-septal reparative fibrosis (prevalent) Parenchymal bands, irregular interfaces, reticular opacities Traction bronchiectasis with or without honeycombing	Final evolution to fibrosis Possible traction bronchiectasis and reticulation	**Clinical manifestations** Low diffusing capacity for carbon monoxide (DLCO), altered gas exchange **Suggested treatment** Symptomatic treatment (i.e., oxygen, corticosteroids, antifibrotic drugs like nintedanib, antibiotics, etc.)

COVID-19, coronavirus disease 2019; ARDS, acute respiratory distress syndrome; DAD, diffuse alveolar damage; NIV, non-invasive ventilation; NIRS, non-invasive respiratory support; HFNC, high-flow nasal cannula; CPAP, continuous positive airway pressure; IMV, invasive mechanical ventilation; V/Q, ventilation/perfusion ratio; FiO_2_, fraction of inspired oxygen; V_T_, tidal volume; PBW, predicted body weight; PEEP, positive end-expiratory pressure; FOT, forced oscillation technique; DLCO, diffusing capacity carbon monoxide.

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
