# Peer review of "Different Methods to Improve the Monitoring of Noninvasive Respiratory Support of Patients with Severe Pneumonia/ARDS Due to COVID-19: An Update"

_jcm, 2022, doi:10.3390/jcm11061704_

Round 1

Reviewer 1 Report

The review by Pelosi et al addresses an issue that is still filling the printed and electronic pages of the media. Taking its cue from the treatment of 'traditional' ARDS, it deals with particular aspects of the management of patients with severe respiratory failure due to SARS-CoV-2, without losing sight of the pathophysiology of the syndrome. It is, in my opinion, a well done and comprehensive work, which highlights the difficulties created by the different ways in which patients are presented. I particularly appreciated the treatment of the importance attributed to the information that can be obtained from the thoracic ultrasound, a rapid, non-invasive, repeatable, reliable means, within the reach of all doctors, which makes it possible, bed-side, to follow in real time and step by step the evolution of the syndrome and the effectiveness of the therapeutic interventions. The iconographic support is exhaustive, the flow-chart in Figure 5 is useful, as is Table 1, which succeeds in summarising the extensive treatment of the subject. The bibliography is adequate and almost free of self-references.

Translated with www.DeepL.com/Translator (free version)

Author Response

We would like to thank the reviewer for all comments and observations

Reviewer 2 Report

The authors in this interesting manuscript describe the need and methods to monitor non-invasive monitoring in patients with COIVD-19 ARDS. Non-invasive ventilation although a great management strategy in patients with early ARDS may not be beneficial to all patients, and may actually harm certain individuals. There is need to evaluate these patients on case by case to determine the best management strategy. 

A few comments and clarifications

  • Can the authors comment on the percentage of Phenotype 1 and 2, can patients then progress form phenotype 1 to 2 and if so what percentage. 
  • Can the authors clarify Phenotype 1 as L and 2 as H early in the manuscript. 
  • With respect to early intubation, it is hard to make a case for early intubation for all Phenotype H patients, definitely agree with early evaluation of these patients for invasive ventilation but a trial of Non invasive ventilation seems to work for a vast majority of these patients.  
  • Most places are not equipped to measure noninvasive compliance by FOT and may be a labor intensive process. 
  • Significant patients with COVID-19 ARDS also have heart failure, renal failure etc which may give the appearance of a phenotype 2 but these patients actually benefit from Non-invasive ventilation. 
  • p-SILI may be an indication for early mechanical ventilation. Patient on NIPPV who are generating massive negative pressures and high tidal volumes in the range of 10-12 cc/kg IBW. Can this be included in the management algorithm.
  •  The description and use of lung ultrasound in the manuscript does not add much value in determining invasive vs non invasive management in these patients and is extremely challenging to use as a monitoring tool. 

Author Response

The authors in this interesting manuscript describe the need and methods to monitor non-invasive monitoring in patients with COIVD-19 ARDS. Non-invasive ventilation although a great management strategy in patients with early ARDS may not be beneficial to all patients, and may actually harm certain individuals. There is need to evaluate these patients on case by case to determine the best management strategy. 

Re: We would like to thank the reviewer and the editor for the positive comments and having given us the possibility to improve our manuscript.

A few comments and clarifications

1) Can the authors comment on the percentage of Phenotype 1 and 2, can patients then progress form phenotype 1 to 2 and if so what percentage.

Re: We agree with the reviewer’s suggestions. In the revised version of the manuscript, we modified the paragraph below and new several references were added:Several studies reported that phenotype L is most frequent (70-80% of patients) than phenotype H (20-30%) [12,21-24]. In their study, Gattinoni et al reported that the transition from phenotype L to type H may be due to both the evolution of COVID-19 pneumonia and the injury attributable to high-stress ventilation. The same group concluded that type L and H are best identified by CT scan, which could be useful also to recognize other possible comorbidities [12]. In line with recent assumptions, the proposed COVID-19-related phenotypes may be considered as the two extremes of a unique evolving disease [21-24]. Recent article demonstrated that phenotype H was more common among females and was associated with a higher risk of ARDS and burden of comorbidities, including diabetes. Furthermore, patients classified as phenotype H were liberated from mechanical ventilation later and characterized by higher mortality as compared to phenotype L patients [12,21-24]”.

2) Can the authors clarify Phenotype 1 as L and 2 as H early in the manuscript. 

Re: As suggested by the reviewer, we better clarified Phenotype 1 as L and 2 as H early in the revised manuscript: “From a clinical point of view, distinct chest CT patterns have been categorized into different COVID-19 phenotypes that can benefit from different clinical management: phenotype 1 (or type L, is characterized by Low elastance, high lung compliance, low ventilation/perfusion ratio, low weight, low recruitability) [21, 22], typical of mild to moderate disease that can initially benefit from noninvasive respiratory support, presents as multiple, often bilateral, focal over-perfused ground glass opacities and normally aerated zones; phenotype 2 (or type H, i.e. High elastance, low lung compliance, high ventilation/perfusion ratio, high lung weight, high recruitability) [21, 22], more typical of the most severe COVID-19, it shows inhomogeneously distributed hyper- and hypo-perfused areas with predominant atelectasis and a patchy ARDS-like pattern [16, 22-24]”.

3) With respect to early intubation, it is hard to make a case for early intubation for all Phenotype H patients, definitely agree with early evaluation of these patients for invasive ventilation but a trial of Non invasive ventilation seems to work for a vast majority of these patients.  

Re: We agree with the reviewer and we better clarified this important issue in the revised version of the manuscript:NIRS represents a possible first step in the management of COVID-19 patients [63]. Several observations reported that patients with COVID-19 phenotype 2 (or H), who are at higher risk of disease progression could be need early intubation, while those who present with phenotype 1 (or L) can be initially managed with NIRS [12,23,24]”.

4) Most places are not equipped to measure noninvasive compliance by FOT and may be a labor intensive process. 

Re: we agree with reviewer’s comments that most places are not equipped with the technique. This issue was better discussed in the revised version of the manuscript in this sentence: “At present time compliance can be noninvasively measured by the Forced Oscillation Technique (FOT) parameter reactance. At present time compliance can be noninvasively measured by the Forced Oscillation Technique (FOT) parameter reactance. FOT is a non-invasive, non-expensive and fast approach which measures lung function using sinusoidal pressure waves generated by a loudspeaker and forced into the lung during tidal breathing, requiring no effort maneuvers [52]. Furthermore, FOT is a simple approach to measure lung mechanics since it needs the patient to have spontaneous tidal volume breathing for maximus 10 breath and it uses a portable instrument. We showed in our preliminary abstract that it is feasible bedside in patients with critical COVID19 pneumonia [22]”.

5) Significant patients with COVID-19 ARDS also have heart failure, renal failure etc which may give the appearance of a phenotype 2 but these patients actually benefit from Non-invasive ventilation. 

Re: We agree with reviewer’s comments and we better discussed this issue in the revised version of the manuscript:In their study, Gattinoni et al reported that the transition from phenotype L to type H may be due to both the evolution of COVID-19 pneumonia and the injury attributable to high-stress ventilation. The same group concluded that type L and H are best identified by CT scan, which could be useful also to recognize other possible comorbidities [12]. In line with recent assumptions, the proposed COVID-19-related phenotypes may be considered as the two extremes of a unique evolving disease [21-24]. Recent article demonstrated that phenotype H was more common among females and was associated with a higher risk of ARDS and burden of comorbidities, including diabetes. Furthermore, patients classified as phenotype H were liberated from mechanical ventilation later and characterized by higher mortality as compared to phenotype L patients [12,21-24]”.

6) p-SILI may be an indication for early mechanical ventilation. Patient on NIPPV who are generating massive negative pressures and high tidal volumes in the range of 10-12 cc/kg IBW. Can this be included in the management algorithm. Re: We agree with reviewer’s comments and we improve the management algorithm as requested.

7)  The description and use of lung ultrasound in the manuscript does not add much value in determining invasive vs non invasive management in these patients and is extremely challenging to use as a monitoring tool. 

Re: In agreement with the first reviewer who “particularly appreciated the importance attributed to the information that can be obtained from the thoracic ultrasound”, we decided to improve the LUS paragraph taking into consideration the important observations from the second reviewer: “Thus, when combined with clinical data, LUS can provide a potent monitoring aid in patients with suspected COVID-19 pneumonia, reflecting and anticipating CT findings. However, LUS does not add much value in determining invasive versus non-invasive management in these patients and it is extremely challenging to use as a monitoring tool. LUS is operator-dependent, and it should be performed and interpreted by an experienced professional. Furthermore, LUS only evaluates pleural and subpleural changes, while deeper lesions could be missed since ultrasound waves cannot penetrate highly aerated lungs. Ultrasound findings also depend on the clinical characteristics of the patient; obesity and subcutaneous emphysema can interfere with passage of US waves and make the assessment more difficult”.

Round 2

Reviewer 2 Report

The changes look acceptable to me.